# Neural Execution Engines

## Abstract

Turing complete computation and reasoning are often regarded as necessary precursors to general intelligence. There has been a significant body of work studying neural networks that mimic general computation, but these networks fail to generalize to data distributions that are outside of their training set. We study this problem through the lens of fundamental computer science problems: sorting and graph processing. We modify the masking mechanism of a transformer in order to allow them to implement rudimentary functions with strong generalization. We call this model the Neural Execution Engine, and show that it learns, through supervision, to numerically compute the basic subroutines comprising these algorithms with near perfect accuracy. Moreover, it retains this level of accuracy while generalizing to unseen data and long sequences outside of the training distribution.

## 1 Introduction

Neural networks are universal function approximators (Hornik et al., 1989), meaning that provided enough data and perfect optimization they should be able to learn arbitrarily complicated functions. In recent years, there have been proposals of neural network architectures that are designed to implement general programs (Graves et al., 2014; Kaiser & Sutskever, 2016; Graves et al., 2016; Kurach et al., 2016), often inspired by concepts found in conventional computer systems, like pointers (Vinyals et al., 2015). However, these neural networks still have difficulty learning complex programs from input/output pairs, in the sense of strong generalization. That is, generalizing to data distributions that do not necessarily correspond to the training distribution, such as longer sequences and new values.

We hypothesize that much of this difficulty stems from a lack of prior structure, and given enough structure in the form of supervision over intermediate program states, we can train networks to faithfully implement different algorithms. We take several basic algorithms (selection sort, merge sort, Dijkstra's algorithm for shortest paths) and express them in terms of a series of subroutines, as a software engineer would. Each subroutine represents a simple function, and can be composed with others to express the algorithm. In this way, we train neural networks to perform relatively simple tasks in a supervised manner, and obtain complex behaviors through composition.

Although each subroutine represents a simple task compared to the full algorithm, this is nevertheless a challenging learning domain for several reasons. First, each subroutine still requires the network to learn a function in such a way that it can strongly generalize outside of its training distribution. Next, as the goal is to learn general computation, the network will operate on raw numbers: taking as input numbers, or distributions over sets of numbers that it may not have even seen in training. Lastly, each subroutine must be performed accurately enough so that composition results in accurate inference over long runs of the program.

Our main contribution is to show that while a model trained on a complex task in an end-to-end fashion may generalize well on in-distribution test data, this does not necessarily lead to strong generalization. However, the same underlying architecture can be made to strongly generalize by introducing minor modifications and more supervision. This provides a starting point for gradually reducing the amount of required supervision and increasing the sizes of the learned subroutines in order to work towards end-to-end learning of complex algorithms with neural networks. Specifically, we leverage the transformer (Vaswani et al., 2017) to learn the subroutines underlying several common yet sophisticated algorithms from input/output execution traces. Our model uses binary number representations for data values, and separates the notion of control (which part of the input to consider) from execution (what to compute) via a conditional masking mechanism. We show that with this,

transformers can learn effective representations for accurately performing fundamental numeric tasks like comparison and addition, and that allowing the transformer to modulate its own mask in subsequent subroutine calls allows it to generalize to runs of the program that greatly exceed the length of the traces it was trained on, resulting in near perfect performance on larger tasks. We refer to these networks over subroutines as neural execution engines (NEEs).

## 2 BACKGROUND

### 2.1 TRANSFORMERS

Transformers are a family of models that represent the current state-of-the-art in sequence learning (Vaswani et al., 2017; Devlin et al., 2018; Radford et al., 2019). In contrast to recurrent networks that process inputs sequentially, transformers process an entire sequence simultaneously using a self-attention mechanism that allows each input token to attend to every other token in the sequence when computing its transformation. This mechanism occurs within an attention block, and several blocks are stacked to form both an encoder to embed an input sequence, and a decoder to produce an output sequence. Here, we will formalize some of the components of the vanilla transformer architecture so that we can outline our modifications in subsequent sections.

Given input token sequences $\mathbf{x}_1, \mathbf{x}_2, \ldots, \mathbf{x}_{L_1} \in \mathcal{X}$ and output token sequences $\mathbf{y}_1, \mathbf{y}_2, \ldots, \mathbf{y}_{L_2} \in \mathcal{Y}$, where $\mathbf{x}_i, \mathbf{y}_j \in \mathbb{Z}^+$, a transformer learns a mapping $\mathcal{X} \to \mathcal{Y}$. First, the tokens are individually embedded to form $\hat{\mathbf{x}}_i, \hat{\mathbf{y}}_j \in \mathbb{R}^d$. The main module of the transformer architecture is the self-attention layer, which outputs a transformation of each vector as a convex combination of values, modulated by the affinity of the vector to every other vector in the sequence. Note that for this paper, we do not use positional encodings (which we found to hurt performance in our tasks), and single-headed attention. Self-attention layers are followed by a position-wise feed-forward neural network layer, forming a self-attention block. These blocks are composed to form the encoder and decoder of the transformer, with the outputs of the encoder being used as queries and keys for the decoder. More details can be found in Vaswani et al. (2017).

An important component for our purposes is the self-attention mask. This is used to prevent certain positions from propagating information to other positions. Typically this is used for decoding, to ensure that the model can only condition on past outputs during sequential generation. In our case, we will consider masking in the encoder as an explicit way for the model to condition on the part of the sequence that it needs at a given point in its computation.

### 2.2 THE SORTING TASK

We use sorting to frame our exploration into the capability of neural networks to mimic general execution. Sorting is a useful abstraction to examine because it is clearly defined at a high level: the sorting task involves mapping a list of unsorted numbers to its sorted counterpart. *Generalization* (performance on unseen sequences) is clearly distinct from *strong generalization* (performance on sequences that are longer that the training distribution). Additionally, there are many different algorithms that computer scientists traditionally use to implement sorting. This provides a rich domain of interest in which we can use supervision to learn different sorting algorithms.

### 2.3 NUMBER REPRESENTATIONS

An essential component of general computation is manipulating numbers (Von Neumann, 1993). Given that our goal is strong generalization, it is also necessary for the number system that we use to work in large ranges and generalize outside of its training domain (as it is not tractable to train the network on all integers). Neural networks generally use either categorical, one-hot, or integer number representations. Prior work has found that scalar numbers have difficulty representing large ranges (Trask et al., 2018) and that binary numbers are a useful representation that generalize outside of their training domain (Kaiser & Sutskever, 2016; Shi et al., 2019), assuming each bit is toggled during training. Binary is a hierarchical format, where each bit extends the training range of the string exponentially, leading to a small bit string that can represent massive number ranges. In this paper, we focus on unsigned integers using the binary representation.

## 3 NEURAL EXECUTION ENGINES

Simply stated, a neural execution engine (NEE) is a transformer that takes as input binary numbers and an encoding mask, and outputs either data values, a pointer, or both. The pointer can optionally be used to modify the mask the next time the NEE is invoked. This architecture is reminiscent of

graph attention networks (GATs) (Veličković et al., 2017), where the graph is used to dictate the encoder mask. An NEE is essentially a GAT that can modify its own graph. In this section, we will describe the specifics behind how we apply this mechanism, before demonstrating its performance on several algorithmic tasks in subsequent sections.

## 3.1 METHODOLOGY

We now describe the key components of the NEE mechanism.

**Bitwise Embeddings.** As input to an NEE, we embed binary vectors using a linear projection. This is equivalent to defining a learnable vector for each bit position, and then summing these vectors elementwise, modulated by the value of their corresponding bit. That is, given an embedding vector $\mathbf{v}_i$ for each bit $i$, for an $n$-bit input vector $\mathbf{x}$, we would compute $\hat{\mathbf{x}} = \sum_{i=1}^{n} x_i \mathbf{v}_i$. For example, $\text{emb}(1001) = v_0 + v_3$.

Two important tokens for our purposes are *start $s$* and *end $e$*. These are commonly used in natural language data to denote the start and end of a sequence. We use $s$ as input to the decoder, and $e$ to denote the end of an input sequence. As we are dealing with numerical computation, we find it convenient to set $s = 0$ and to have the model learn an embedding vector for $e$ such that $e = \infty$. That is, the model will learn $e > x$ for all $x \neq e$ and that $e + x = e$.

**Conditional Masking.** The encoder of a transformer takes as input both a set of values and a mask, which is used to force the encoder to ignore certain inputs. *In the NEE mechanism, we use the output pointer of the decoder to modify the mask for a subsequent call of the encoder.* In this way, the inputs represent a memory state, and the mask represents a set of pointers into the memory. NEE effectively learns where to focus its attention for performing computation.

A mask $\mathbf{M}$ is a binary matrix, where a value of $0$ indicates that the input should be considered, and $1$ indicates that it should be ignored. The encoder mask is used to zero out attention weights of ignored input numbers. Given a binary vector $\mathbf{b}$, a mask is obtained by broadcasting $\mathbf{b}$ along the sequence length dimension and other higher dimensions. The value $b_i = 0$ would indicate that the $i^{\text{th}}$ input should be considered, and $b_i = 1$ would indicate that the $i^{\text{th}}$ input should be ignored. We will refer to $\mathbf{b}$ as the mask vector.

Given an input mask vector $\mathbf{b}$, we need to compute the updated mask vector $\hat{\mathbf{b}}$ to serve as the next input mask. As most entries of the mask do not change with each iteration, we output only the information required to update the mask. Practically, this means a one-hot encoding $\mathbf{o}$ representing the position of interest to mask out. We then compute the updated mask $\hat{\mathbf{b}}$ using a bitwise XOR function, $\hat{\mathbf{b}} = \text{XOR}(\mathbf{b}, \mathbf{o})$. This is sufficient for the subroutines considered in this paper, but the transformer can output arbitrary masks, and future work will focus on allowing the network to make more complex updates.

Using this mechanism, we can train NEE to mimic the behavior of a variety of subroutines by training it on execution traces using teacher forcing (Williams & Zipser, 1989). At inference, we simply choose the argmax of the pointer output head.

**Encoding Independent Inputs.** We will show examples where the function must operate on a sequence of inputs, but where subsets of the inputs can be treated independently. Specifically, an elementwise sum or min between pairs of numbers. For transformers, this can be done by an explicit *reshape* operation that turns the sequence into a mini-batch, enabling parallel processing on the entire batch. A subsequent reshape can turn the output back into a sequence.

**Encoding Dependent Inputs.** When we receive a sequence of inputs where subsequences are dependent, we concatenate the subsequences and delineate them using the $e$ token. For example, if we had two subsequences $[a, b]$ and $[c, d]$, we would input $[a, b, e, c, d, e]$. For the corresponding mask, we could give $[0, 0, 1, 0, 0, 1]$ to consider all tokens, or $[0, 1, 1, 0, 1, 1]$ if we intend to iterate over the subsequences in order. If required, NEE will output a pointer, and we can use this to update the mask as needed. This mechanism will be useful in merge sort, where a fundamental subroutine is merging two sorted lists into a single sorted list.

**Decoding.** Decoding in a NEE involves giving the start token $s$ to the decoder and using it to compute attention blocks over the encoder outputs. In this case, we only consider outputting a single value and/or pointer, as opposed to a sequence.

## 4 LEARNING TO SORT

As we mentioned in Section 2, we use sorting to frame our exploration into the capability of neural networks to mimic general execution.

### 4.1 A SEQUENCE TO SEQUENCE VIEW

We first study how well the vanilla transformer (Section 2.1) learns to sort. We model it in a conventional seq2seq fashion (Sutskever et al., 2014) with input examples of unsorted sequences (length $\leq L$) of $n$-bit binary numbers ($L$=8 and $n$=8) and output examples of the correctly sorted sequence. The inputs to the encoder and decoder are sequences of bitwise embedded vectors (Section 2.3). The decoder uses a greedy decoding strategy. The

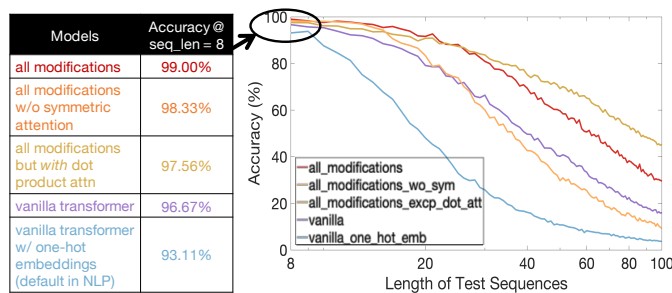

| Models | Accuracy @ seq_len = 8 |
|---|---|
| all modifications | 99.00% |
| all modifications w/o symmetric attention | 98.33% |
| all modifications but *with* dot product attn | 97.56% |
| vanilla transformer | 96.67% |
| vanilla transformer w/ one-hot embeddings (default in NLP) | 93.11% |

Figure 1: Seq2Seq sorting performance of transformer variants trained on sequences of up to length 8.

outputs of the decoder are one-hot 256-dimensional vectors representing the unnormalized log-probabilities of the output numbers. We find that randomly generated numbers are easier to sort than numbers that differ in a small number(e.g., 1 vs. 2, 53 vs. 54). Thus, we include both examples in the training distribution (70% random numbers, 30% numbers with small differences). [1] The performance of this vanilla transformer, evaluated as achieving an exact content and positional match to the correct output example, is shown in Figure 1 (where the test distribution consists of 60% random numbers and 40% numbers with small differences).

To increase performance, we make a number of modifications to better tune the encoder to process binary numbers. These modifications are driven by the unsatisfactory performance of the vanilla transformer in distinguishing close numbers (Appendix A.2, Figure 11c). To aid in the ability of the number to learn numerical similarity in carry-out cases when many bits flip (32 vs. 31, 64 vs. 63), we use bitwise embeddings and replace the dot product self-attention with a symmetric feed-forward self-attention (Figure 11c). To help handle small differences in general, we use shared instead of independent linear projections of queries, keys and values. To aid in reconstructing small numerical differences, we scale up the residual connection strength by a factor of $1.5$. These modifications over the original transformer model are shown in the Appendix (Figure 10b).

Figure 1 shows the sorting performance of five models: (1) the model with all the modifications mentioned above, (2) the model with all modifications but with non-symmetric attention, (3) the model with all modifications but with dot-product attention (instead of symmetric feed-forward self-attention), (4) the vanilla transformer model with bitwise embedding, and (5) the vanilla transformer model with one-hot embedding (which is default in NLP). The results of more variants and data mixes are shown in the Appendix (Figure 11). We observe that while the transformer variants all have reasonably high performance in sorting lists of numbers that are $\leq 8$ elements long from input/output examples, the models fail to generalize to longer sequences, and performance sharply drops as the sequence length increases.

To understand why performance degrades as the test sequences get longer, we plot the attention matrix of the last layer in the decoder (Figure 2a). The model accurately attends to the first few numbers in the sequence (distinct dots in the chart) but the attention distribution becomes "fuzzy" as the sequence length increases beyond 8 numbers, often resulting in the same number being repeatedly predicted. Next, we seek to investigate the following question: Can we retain high attention resolution (and corresponding model accuracy) by restricting the model to only observe the first scenario repeatedly?

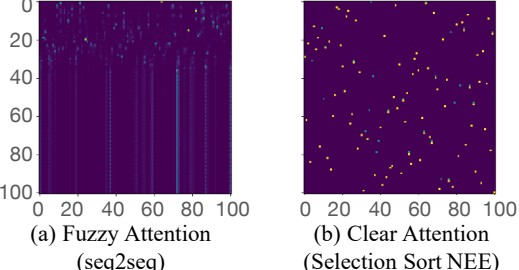

(a) Fuzzy Attention (seq2seq)   (b) Clear Attention (Selection Sort NEE)

Figure 2: Visualizing the decoder attention weights of sorting transformers. Attention is over each row. The attention in the seq2seq transformer saturates as the output sequence length increases, while NEE maintains sharp attention weights.

---

[1]Throughout this work, the preponderance of errors are that regenerated numbers are off by small differences.

## 4.2 SUBROUTINE LEVEL TRACES: SELECTION SORT

Using this insight, we break the problem up into pieces and increase the amount of supervision provided during training. Instead of learning to perform the entire computation from only input/output examples, we pick a simple sorting algorithm to emulate (selection sort in Figure 3) and train the transformer to learn one action that can be composed in a loop to solve the task—much as a programmer would while writing code. Our guide for selecting these subroutines to train an NEE on is to determine the largest source code function to learn, where the output of the function is data-dependent. That is, static parameters (like the XOR or reshape in Section 4.3) are not learned. For the case of selection sort, the task is to find the minimum of the current list.

Since the transformer had difficulty clearly attending to values beyond the training sequence length, we restrict the locations in the unsorted sequence where the transformer could possibly attend in every iteration. This is accomplished by producing a conditional mask to mask out the data elements that have already been appended to the sorted_list and feeding that mask back into the transformer (shown on the right side of Figure 3). This modification separates the control from the computation (which elements should

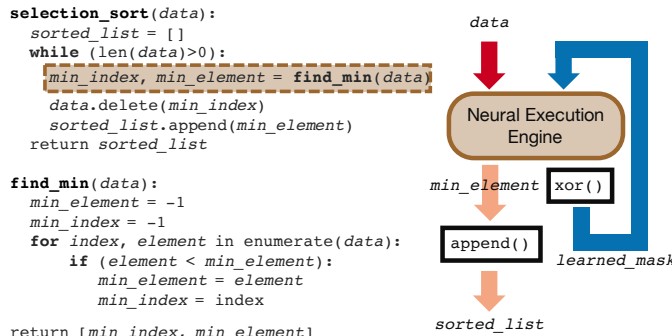

Figure 3: Selection sort code translated to NEE architecture. The colored box (find_min) is learned from input/output examples, but one potential code implementation is shown.

be considered) from the computation itself (find the minimum value of the list). This allows the transformer to learn output logits of much larger magnitude, resulting in sharper attention, as shown in Figure 2b. Our experimental results consequently demonstrate strong generalization, perfectly sorting sequences of up to length 100, as shown in Figure 12a (Appendix A.2).

## 4.3 RECURSIVE SUBROUTINES: MERGE SORT

Given the success of an NEE repeatedly performing an action (finding the minimum of a list) in order to implement selection sort, we explore how well the approach generalizes to other sorting algorithms. As a programmer could write smaller subroutines to accomplish the sorting task in many different ways, NEEs should be able mimic these subroutines from execution level traces. We therefore select a very different sorting algorithm for study, merge sort. The code for one implementation of merge-sort is shown in Figure 4.

This code is broadly broken up into two subroutines, data decomposition (merge_sort) and an action (merge). Every call to merge_sort divides the list in half until there is one element left, which by definition is already sorted. Then, merge unrolls the recursive tree, combining every 2 elements (then every 4, 8, etc.) until the list is fully sorted. Recursive algorithms like merge sort generally consist of these two steps (the "recursive case" and the "base case").

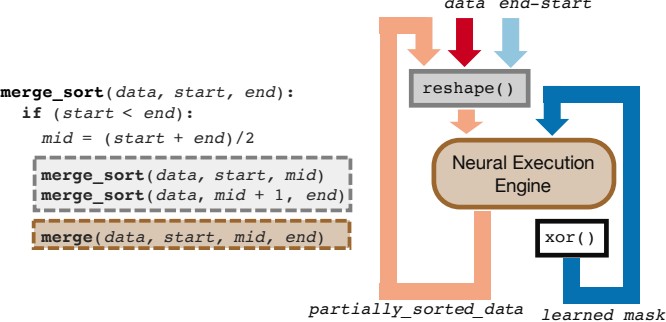

Figure 4: Merge sort code translated to NEE architecture.

The recursive case is commonly based around data movement, and we use a reshape() operation to divide or combine lists every time and emulate the start and mid pointers in the application (this is a static parameter). It is possible to use the learned mask of the transformer to learn the recursive decomposition serially (e.g., have the mask shift between the first pair of numbers, then the second

pair, etc.), but the reshape function allows us to teach the transformer to swap all pairs of data for every level in the tree in parallel.

More concretely, the transformer in Figure 4 implements the computation described in Figure 5. The mask controls which numbers are seen by the encoder (1 meaning masked). Every timestep, the model outputs the smallest number from the unmasked numbers and the position of the currently selected number. This is used to produce the next mask with a SHIFT and a XOR. We do not learn these operations since they are static and don't depend on data. The model stops when it outputs an end token. Figure 5 shows the computation for one leaf of the recursive decomposition, but the reshape allows the transformer to compute on all leaf nodes at a given level in the tree in parallel. Given this model, Figure 6 demonstrates that the NEE is able to learn merge sort with perfect strong generalization over long sequences (up to length 100). The NEE was trained to perform the computation underlying merge sort on sequences of length $\leq 8$.

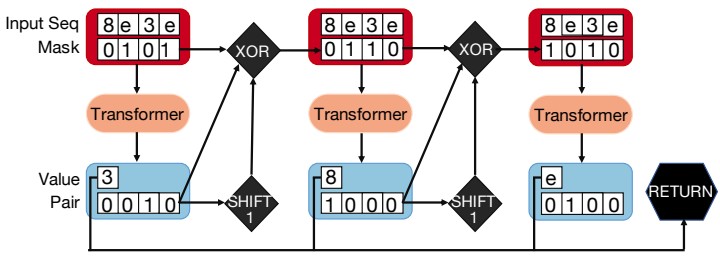

Figure 5: NEE merge sort trace.

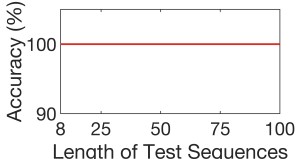

Figure 6: NEE merge sort: strong generalization over long sequences. Training involved sequences of length $\leq 8$.

## 5 COMPOSABLE SUBROUTINES: SHORTEST PATH

While both merge sort and selection sort demonstrated that an NEE can compose subroutines repeatedly to sort a list with perfect accuracy, programmers often need to compose multiple different subroutines to achieve larger goals. In this section, we study whether multiple NEEs can be composed to execute a more complicated algorithm.

To that end, we study a graph algorithm, Dijkstra's algorithm to find shortest paths, shown on the left of Figure 7, which is mirrored in the source code on the left. The algorithm consists of four major steps: (1) Initialization: set the distance from the source node to the other nodes to infinity (*end e* in Section 3.1), then append them into a queue structure for processing; (2) Compute newly found path from the source node to all neighbors of the selected node; (3) Update path lengths if they are smaller than the stored lengths; (4) Select he node with the smallest distance to the source node and remove it from the queue. The algorithm repeats steps (2)–(4) as long as there are elements in the queue.

Computing Dijkstra's algorithm requires the NEEs to learn the three corresponding subroutines, as shown on the right in Figure 7. Finding the minimum between the possible_paths and shortest_path as well as the minimum current shortest_path can be accomplished by the modified transformer trained to accomplish the same

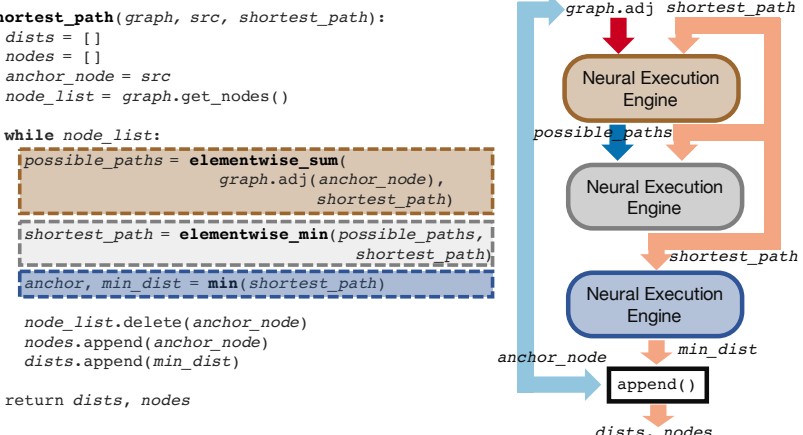

Figure 7: Dijkstra code translated to NEE architecture.

goal for sorting in Section 4.2. The new challenges are to learn addition (Section 5.1) and to compose these learned models sequentially to achieve shortest path in a generalizable way (Section 5.2).

## 5.1 LEARNING ADDITION

The addition subroutine in Dijkstra's algorithm iteratively adds all of the elements in the adjacency list of a selected node to the shortest path that is currently known. To learn this function, we train an NEE to learn the sum of two numbers, then apply it to Dijkstra's algorithm.

Given the input/output examples for this subroutine, we find that addition is a more challenging task than sorting. Zero and infinity (end token, Sec. 2.3) are important concepts for the model to learn. Our bitwise embedding already represents zero as the zero vector. We find it useful to embed infinity as a directly learnable vector. We use a moving average on the parameters of the neural network in evaluation to reduce the noise in weights (Kushner & Yang, 1995; Polyak & Juditsky, 1992), and with this, our modified transformer architecture is capable of learning addition to 100% accuracy.

To understand the number system that the NEE has learned, we visualize the structure of the learned embeddings using PCA to 3 dimensions, and compare the embeddings learned from addition and sorting, shown in Figure 8a and Figure 8b, respectively. Each node is colored based on the number it represents. We find that a highly structured number system has been learned, and the addition embeddings are rigorously arranged such that the sequence of numbers that increase by 1 are placed on a line (shown with arrows in Figure 8a). The sorting embeddings exhibit many small clusters and the numbers in each cluster are related. More detailed visualizations are provided in Appendix A.3.

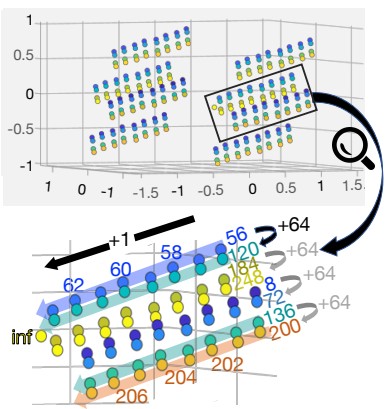

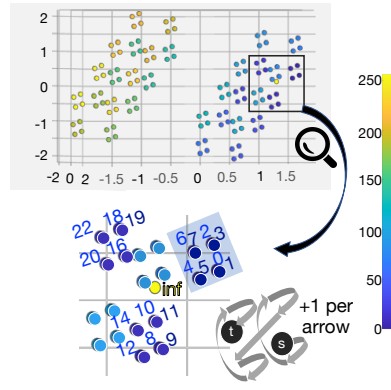

(a) Addition: The embeddings of a sequence of numbers that increase by 1 are placed on a line.

(b) Sorting: The numbers per cluster are related according to the pattern of gray arrows (bottom).

Figure 8: Visualization of learned addition and sorting embeddings.

In addition to testing on unseen *pairs* of numbers in the addition task, we can also test on *completely unseen numbers* to further test the generalization of our model. As shown in Table 1, even with half the data range held out of training, the model achieves high accuracy. This study corroborates prior work that shows that the model only needs to observe the bit being toggled to generalize to the numerical range represented by any given bit (Shi et al., 2019). This is a promising result as it suggests we may be able to extend the framework to much larger bit vectors, where observing every number in training is intractable.

Table 1: Performance for different hold-out % of 8-bit number ranges.

| Hold-out % | 6.25% | 12.5% | 25% | 50% |
|---|---|---|---|---|
| Accuracy | 99.96% | 100% | 99.2% | 98.72% |

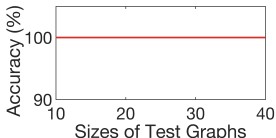

Figure 9: NEE Dijkstra: Training on graphs with ≤ 10 nodes generalizes perfectly to ≤ 40-node graphs.

## 5.2 RESULTS

Given that NEEs can be trained to perform the necessary subroutines underlying Dijkstra's algorithm, we demonstrate that they can then be composed to reliably compute the overall algorithm and generalize to much larger graphs. A step-by-step view of the operations that the NEEs perform are shown in Figure 15 in Appendix A.4. The examples are Erdős-Rényi random graphs (since the types of graphs do not matter). We train on graphs with up to 8 nodes and test on graphs up to 40 nodes, with 100 graphs evaluated at each size. Weights are randomly assigned within the allowed 8-bit number range. We evaluate the prediction accuracy on the final output (the shortest path of all nodes to the source nodes) and achieve 100% test accuracy with graph sizes up to 40 nodes (Figure 9).

## 6 RELATED WORK

There have been a number of recent proposals for neural networks that learn to perform computation that are inspired by computer architectures (Graves et al., 2014; 2016; Kaiser & Sutskever, 2016; Kurach et al., 2016; Vinyals et al., 2015). These attempt to learn complex algorithms purely from weak supervision, i.e., input/output pairs. Through universality, these architectures are able to theoretically represent any computable function, however practically they tend to have issues generalizing to longer sequences. They are typically trained on scalar data values in limited ranges, or focus purely on pointer arithmetic. Our focus is on learning simple, composable functions on raw numerical values in a way that will generalize to extremely long sequences that have not been seen during training. Furthermore, instead of developing a custom architecture, we focus on using transformers (Vaswani et al., 2017) with very little modification, leveraging the encoding mask to allow the transformer to focus its capacity on relevant input locations.

Neural program synthesis (Neelakantan et al., 2016; Reed & De Freitas, 2016; Parisotto et al., 2017; Devlin et al., 2017; Cai et al., 2017; Bunel et al., 2018) uses neural networks to generate programs that are capable of solving tasks represented by input/output pairs, with the goal of finding a "correct" program such that it will generalize beyond the training distribution. Related to this area, there is a body of research at the intersection of logic programming and neural networks (Evans & Grefenstette, 2018; Dong et al., 2019), that uses neural networks to learn a set of rules entailing only positive examples from a set of positive and negative examples. Inputs are treated as separate objects, and specific rules are given to the system for the neural network to use. For example, in learning to sort, Dong et al. (2019) give their system all of the numbers involved as one-hot encoded vectors, along their ground-truth comparative relationships, whereas we focus on learning these relationships from binary numbers. Similar to our own work, Reed & De Freitas (2016); Shi et al. (2019); Cai et al. (2017) use execution traces as a form of stronger supervision. Cai et al. (2017) uses execution traces with tail recursion (where the subroutines call themselves), albeit for program synthesis, and show that this leads to improved generalization.

There have been several works on using neural networks to learn number systems for performing arithmetic. Paccanaro & Hinton (2001) directly embeds integers in the range $[-10, 10]$ as vectors and trains these, along with matrices representing relationships between objects. They train these on triplets involving two objects and a relation, to perform modular arithmetic, and show that these embeddings and relationships are capable of learning a coherent structure and generalizing to new triplets. Sutskever & Hinton (2009) expand on this idea, modeling objects as matrices as well, so that relationships can equivalently be treated as objects, allowing the system to learn higher-order relationships. Trask et al. (2018) explores the (poor) generalization capability of neural networks on scalar-values inputs outside of their training range, and develops new architectures that are better suited to scalar arithmetic, improving extrapolation. Several papers have used neural networks to learn binary arithmetic with varying degrees of success (Joulin & Mikolov, 2015; Kaiser & Sutskever, 2016). Recently, Shi et al. (2019) showed that graph neural networks are capable of learning directly from 64-bit binary memory states from execution traces of assembly code, and that this representation generalizes better when predicting program behavior outside of the training range. Finally, Wallace et al. (2019) explored the ability of various language models to perform numerical reasoning after being trained on a question/answer dataset; they found that language models exhibit a high degree of numeracy, especially when using character-level embeddings.

## 7 CONCLUSION

We propose neural execution engines (NEEs), which leverage a learned mask and supervised execution traces to mimic the functionality of subroutines. We demonstrate that while algorithms like sorting are challenging to generalizably learn from input/output examples, we can identify smaller, simpler subroutines that transformers can learn with near-perfect strong generalization. While the functionality of these subroutines is currently limited, future work can expand the complexity of each subroutine that an NEE learns, getting closer to end-to-end learning of complex algorithms with neural networks. There are many natural extensions, including: extending the usage of binary to the transformer outputs, teaching learned masks more complex pointer primitives, using reinforcement learning to replace the supervision provided by teacher forcing, linking the generation of these models to source code, and exploring the link between the learned subroutines of NEE-like architectures and conventional von-Neumann computers, which execute individually encoded instructions sequentially (Von Neumann, 1993).

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

## A APPENDIX

### A.1 HYPERPARAMETERS

For sorting, we found it sufficient to use bitwise embeddings of dimension $d = 16$. For addition, which is a more difficult task, we found it necessary to increase this to $d = 24$. We use no positional encoding for the sorting task, and single-headed attention for all tasks. In the addition task, we use an exponential moving average of the parameters during learning under a two time-scale stochastic approximation (Polyak & Juditsky, 1992) for additional stability in the learned model. The remaining transformer hyperparameters, aside from the changes described next, are set to their defaults.

### A.2 SORTING ABLATIONS

In this section, we show the result of our architecture modifications to performance in learning to sort. We study two scenarios: the first is a seq2seq setup, where the NEE must learn to sort the entire sequence in one encoding/greedy decoding step (making it equivalent to a transformer). The second is when implementing the selection sort algorithm, where the NEE must output the min value and location, and where the updated mask based on this is fed back into the NEE for subsequent iterations, until the $e$ token is returned. Note that the second scenario has much more supervision than the first scenario, which must learn an entire sorting algorithm internally.

Within each scenario, we study 3 different data distributions: the first is where we train on purely random sequences with tokens in $[0, 255] \cup \{e\}$. The second is a mixed setting, where $60\%$ of the examples are drawn randomly, and $40\%$ are drawn from a more difficult distribution, where the numbers are closer in value. The third is a purely difficult setting, where all sequences have numbers that are close to each other in value.

We ablate specific architectural changes in these settings. The original and modified encoder are represented visually in in Figure 10. Specifically, the architectural choices we test are:

- Scaling up the strength of the residual connections by a factor of $1.5$.
- Using an MLP-based attention module (Bahdanau et al., 2015).
- Symmetrizing the MLP-based attention by flipping the order of the inputs and averaging the resulting logit values.
- Using the standard scaled dot product attention.
- Using a binary encoding of input values.
- Using a one-hot encoding of the input values.
- Using a binary encoding as the input, but without any linear embedding.
- Sharing the bitwise embedding projection between the query, key, and value in the attention mechanism.

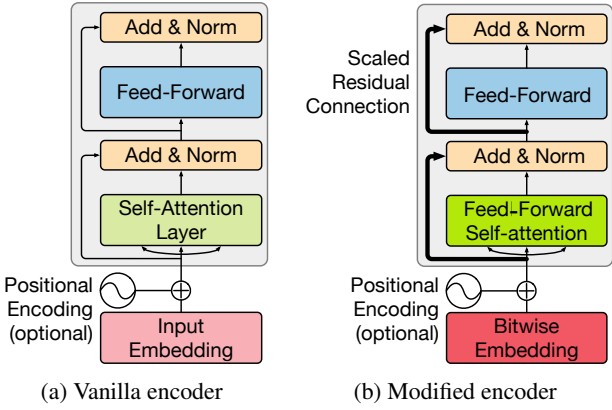

(a) Vanilla encoder      (b) Modified encoder

Figure 10: Baseline transformer (a) and modified transformer (b).

The test accuracy, measured as getting all values and their positions correct, on sequences of length 8 in the *mixed* setting is shown in Table 2. We can see that the architectural changes help improve performance on these sequences up to near-perfect accuracy.

Table 2: Seq2Seq performance for transformer variants at training length of 8 on mixed test sets.

| Models | Accuracy @ seq_len = 8 |
|---|---|
| all_modifications | 99.00% |
| all_modifications_wo_sym | 98.33% |
| all_modifications_excp_res | 95.89% |
| all_modifications_excp_shared_proj | 98.44% |
| all_modifications_excp_dot_att | 97.56% |
| all_modifications_one_hot_emb | 89.56% |
| all_modifications_binary_emb | 84.78% |
| vanilla | 96.67% |
| Vanilla_binary_emb | 77.11% |
| Vanilla_one_hot_emb | 93.11% |

In Figure 11, we show the strong generalization performance of the different architectures. While some changes are able to improve performance in this regime, the performance ultimately drops steeply as the length of the test sequence increases. This is consistent across all test scenarios.

We next run the same ablation, but this time with selection sort using NEEs (of different variants) to perform the find_min operation, and with feedback via conditional masking; the results are shown in Figure 12. In this case, most architectural choices perform well, and strongly generalize to much longer sequences than they were trained on. This shows that NEE is quite robust to the choice of architecture. The exceptions are using raw binary inputs, showing that a bitwise embedding is important for encoding number similarity. Note that the raw bits as inputs still work much better than it does in seq2seq setting regardless in short or long sequences.

Here we list out some random and hard examples as well as the corresponding output (containing some errors) from the vanilla transformer with one-hot encoded input numbers (each number has an independent embedding), which is commonly used in natural language models. The symbol e represents the end token. It can be seen that the model makes more mistakes in hard examples.

Random examples:
100 62 114 66 241 1 63 237 e
181 52 71 254 246 145 118 28 e

Output from Vanilla_one_hot_emb:
1 62 63 66 100 114 237 253 e
28 52 71 118 145 181 246 254 e

Hard examples:
132 126 131 129 127 130 128 125 e
238 239 241 240 243 237 242 244 e

Output from Vanilla_one_hot_emb:
125 126 127 128 129 130 132 e e
237 238 240 244 243 e 237 242 e

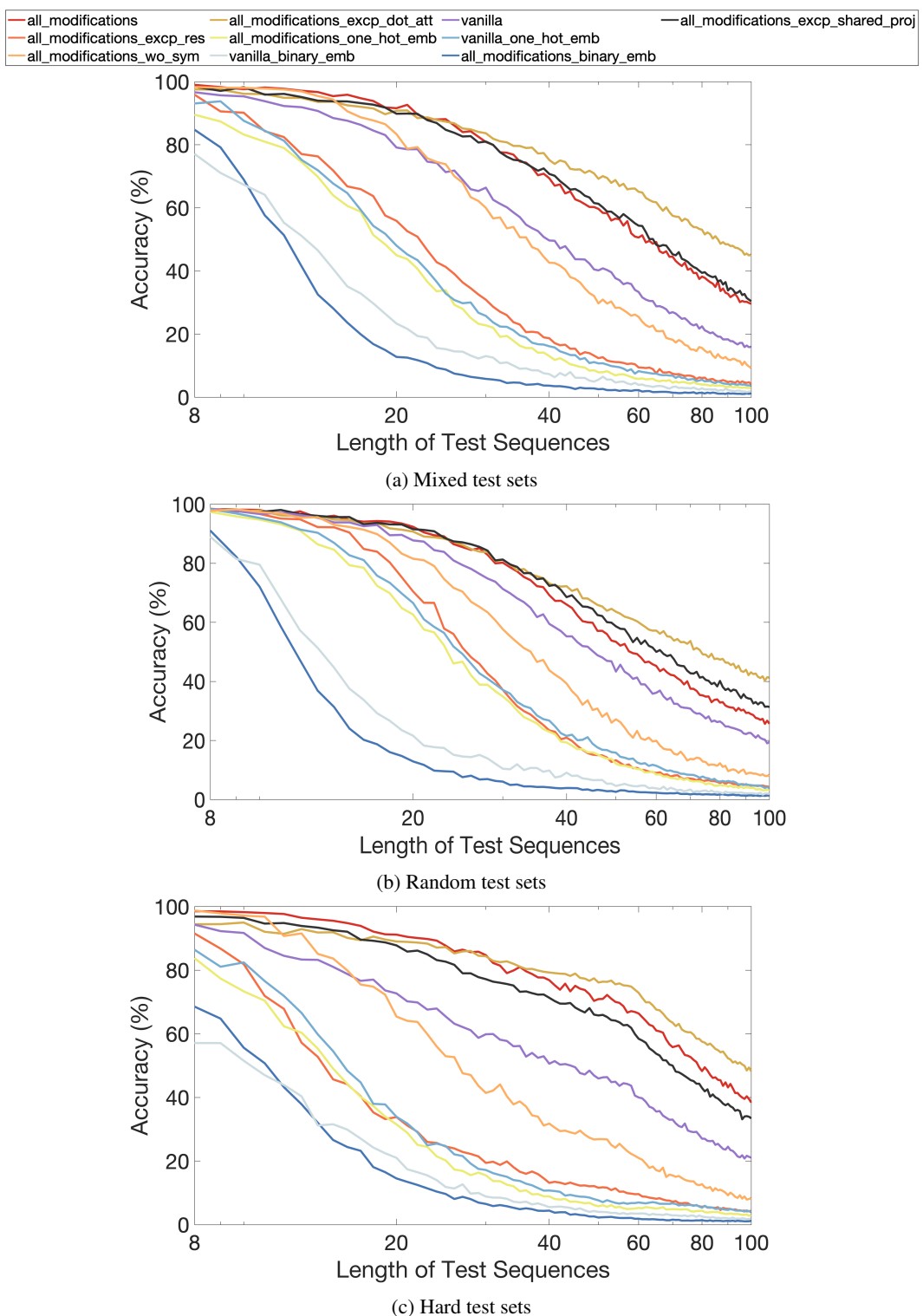

Figure 11: Seq2Seq strong generalization performance (without mask feedback to encoder) on (a) mixed test sets, where test sets consist of 60% random examples and 40% hard examples where the numbers are close to each other. (b) Random test sets. (c) hard test sets, where test sets consist of 100% hard examples where the numbers are close to each other. All models trained on sequences ≤ 8 and tested up to length 100. Vanilla corresponds to the original transformer, with bitwise embeddings.

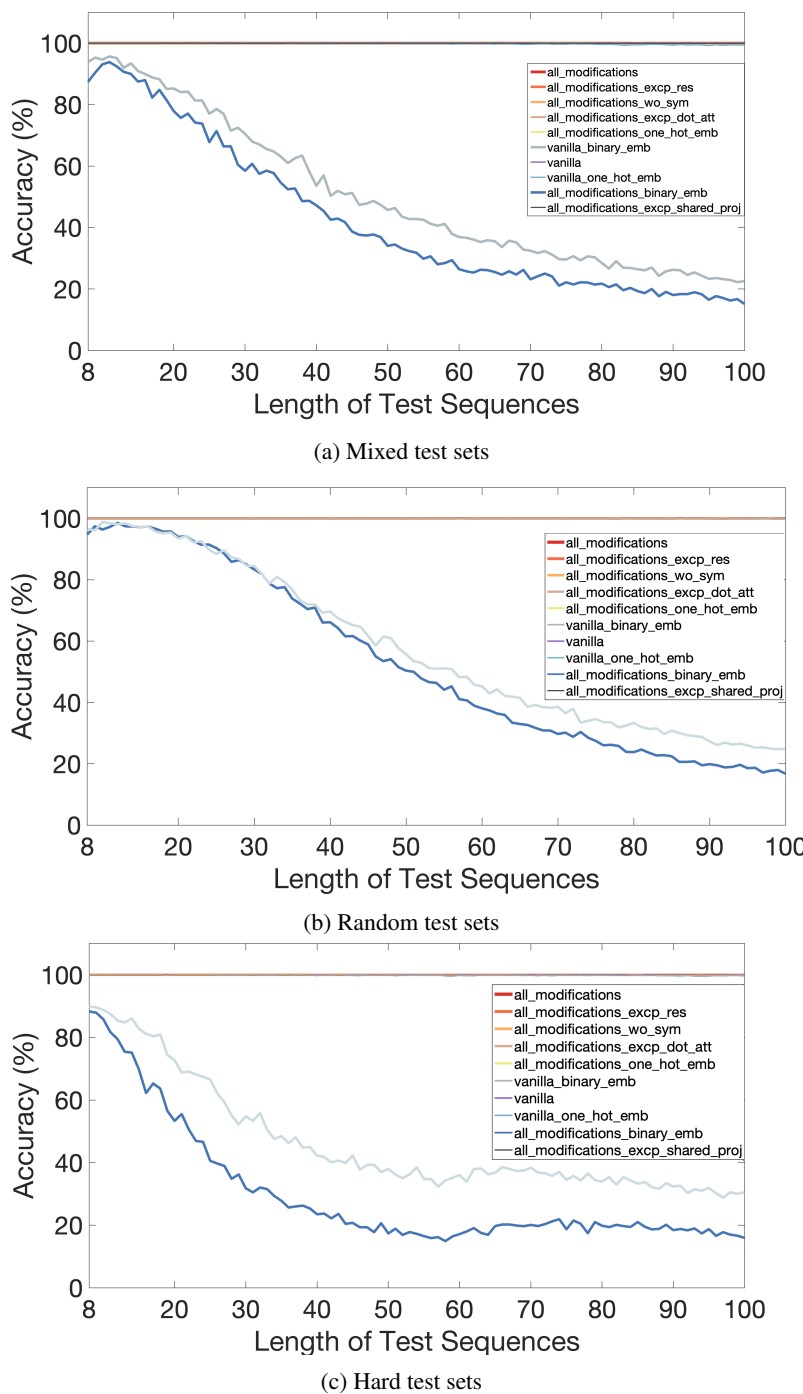

(a) Mixed test sets

(b) Random test sets

(c) Hard test sets

Figure 12: Selection sort strong generalization performance (with mask feedback to the encoder) on (a) mixed test sets, where test sets consist of 60% random examples and 40% hard examples where the numbers are close to each other. (b) Random test sets. (c) hard test sets, where test sets consist of 100% hard examples where the numbers are close to each other. All models trained on sequences $\leq 8$ and tested up to length 100. Vanilla corresponds to the original transformer, with a single modification.

## A.3 DETAILED VISUALIZATION OF LEARNED NUMBER EMBEDDINGS

In Figure 13 we show more detailed visualizations of the learned bitwise embeddings. These are 3-dimensional PCA projections of the full embedding matrix, capturing approximately 75% of the total variance. The main takeaway is that the network is able to learn a coherent number system with a great deal of structure, and that this structure varies depending on the specific task of the network. This is reminiscent of Paccanaro & Hinton (2001), where linear embeddings learned the correct structure to solve a simple modular arithmetic task. Also of note is that the network learns to embed $e$, correspond to infinity, outside of this structure.

Perhaps the most interesting feature is found in Figure 13c. Here, we trained the network on the addition task, however we randomly held out 50% of the input numbers, shown in red. The network still learns a coherent embedding structure, and places the held-out numbers in their correct positions. This is an important feature if the network is to generalize to larger numbers. In particular, we hope to scale NEE to operate on 64-bit values, where seeing each integer at training time is impossible.

Future work will also investigate the resulting embedding from a NEE that performs multiple or more complex tasks.

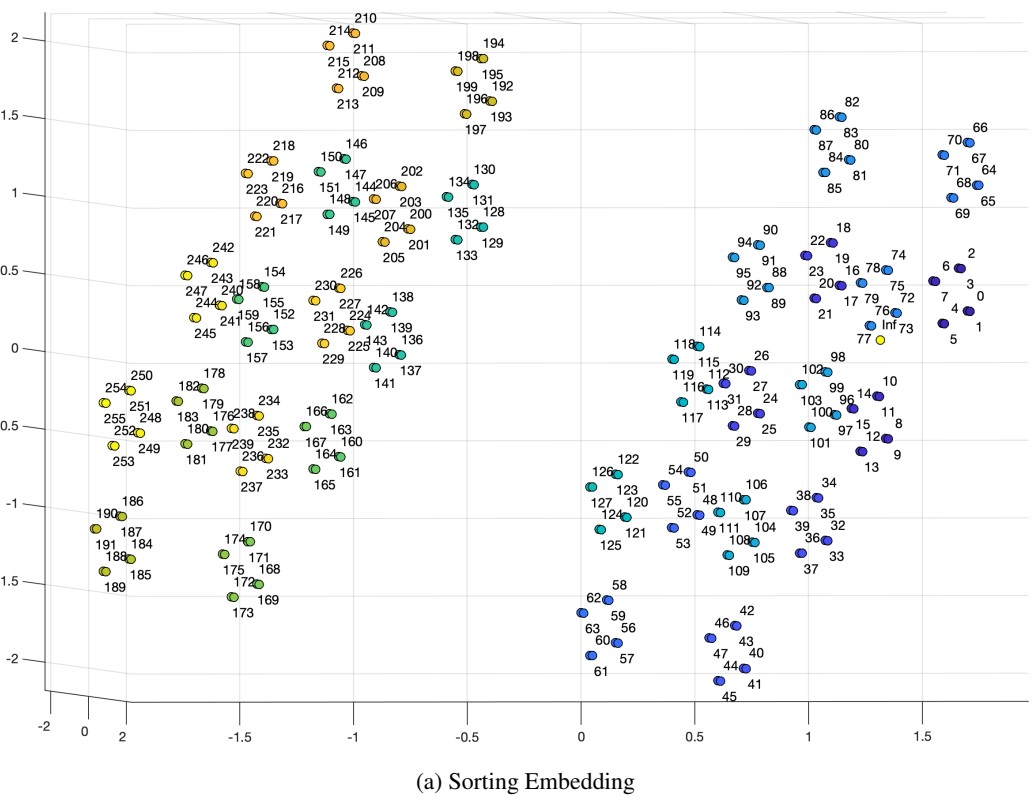

(a) Sorting Embedding

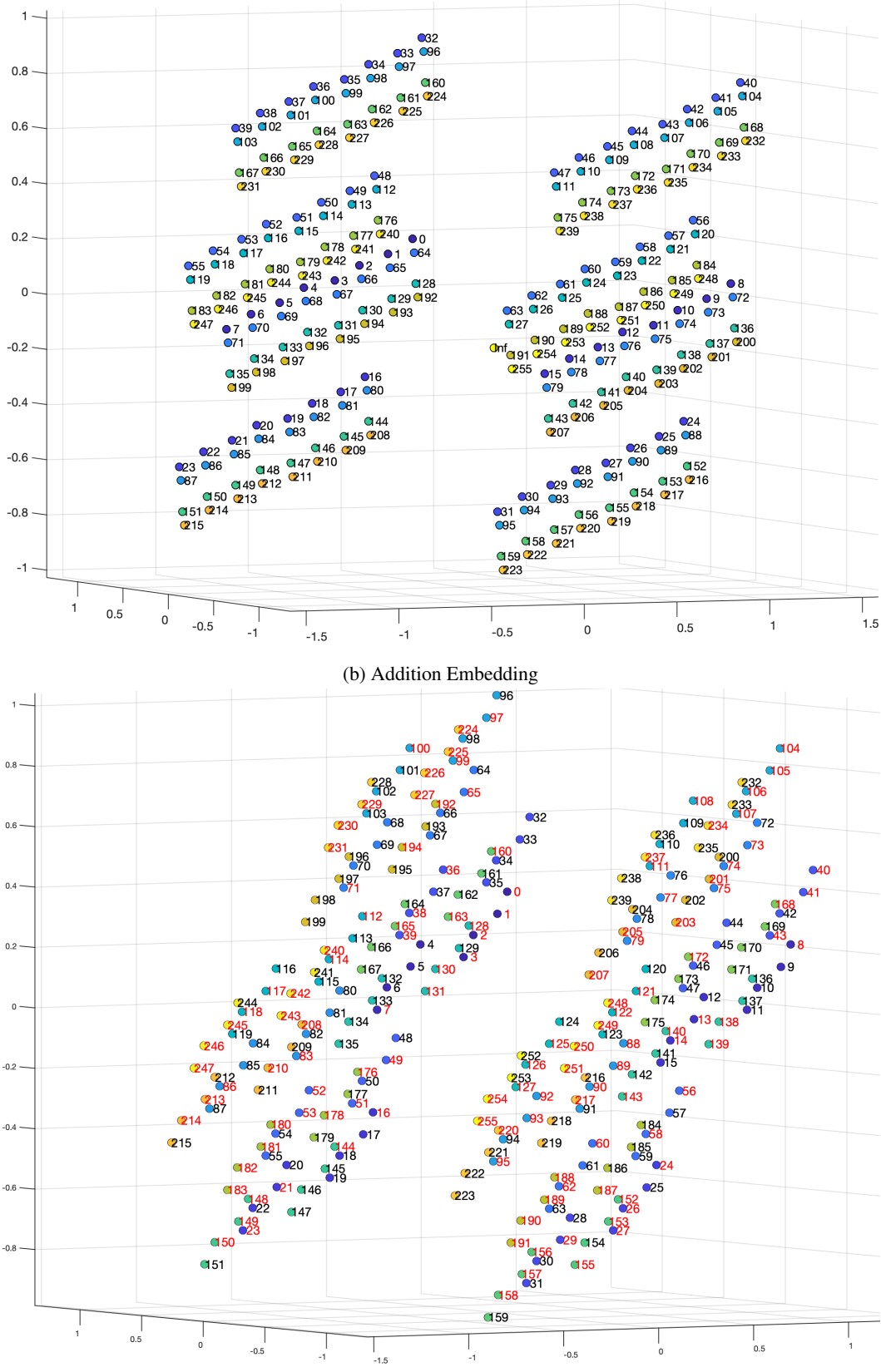

(b) Addition Embedding

(c) Generalization on Addition (numbers randomly held out of training colored red).

Figure 13: 3-dimensional PCA projections of learned bitwise embeddings for (a) sorting, (b) addition, and (c) addition with 50% of the numbers withheld from training.

A.4 DIJKSTRA'S ALGORITHM PERFORMANCE AND EXECUTION TRACE

In Figure 15 we show an execution trace of Dijkstra's algorithm with NEE components for the toy graph shown in Figure 14, where we choose "A" to be the source node. Dijkstra's algorithm uses the weighted adjacency matrix, and maintains in memory the current shortest path values from each node to the source, `shortest_path`, along with the currently considered node that we denote as the "current node" or `anchor`. NEE also maintains a variable `mask` that marks which nodes have already been considered. Given $N$ nodes, and beginning with the source node, Dijkstra's algorithm can be broken up into 3 distinct phases (after initialization), with each one using a NEE to perform the underlying computation.

1. Reference the row of the adjacency matrix indexed by the current node, `dists_current_node`, elementwise add this to the current shortest path values
   `possible_paths = dists_current_node + shortest_path`

2. Take the new shortest path as the elementwise min between `shortest_path` and `possible_paths`

   $$\text{shortest\_path} = \text{elementwise\_min}(\text{shortest\_path}, \text{possible\_paths})$$

3. Select the new "current" node from among those not masked out, and update the mask.
   $\text{anchor} = \text{elementwise\_argmin}(\text{shortest\_path}[\text{mask} \neq 1])$
   $\text{mask} = \text{XOR}(\text{mask}, \text{one\_hot}(\text{anchor}))$

4. If $\sum_i \text{mask}_i = N$ then terminate, otherwise go back to step 1.

Putting this all together, we find that training NEE components to execute the 3 phases on graphs of up to 8 nodes in size generalizes perfectly to graphs of up to 40 nodes (Figure 9).

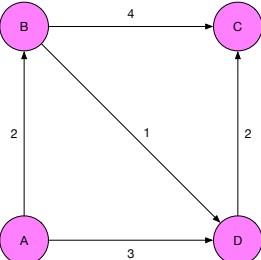

Figure 14: Graph used for shortest path execution trace in Figure 15. Node "A" is used as the source.

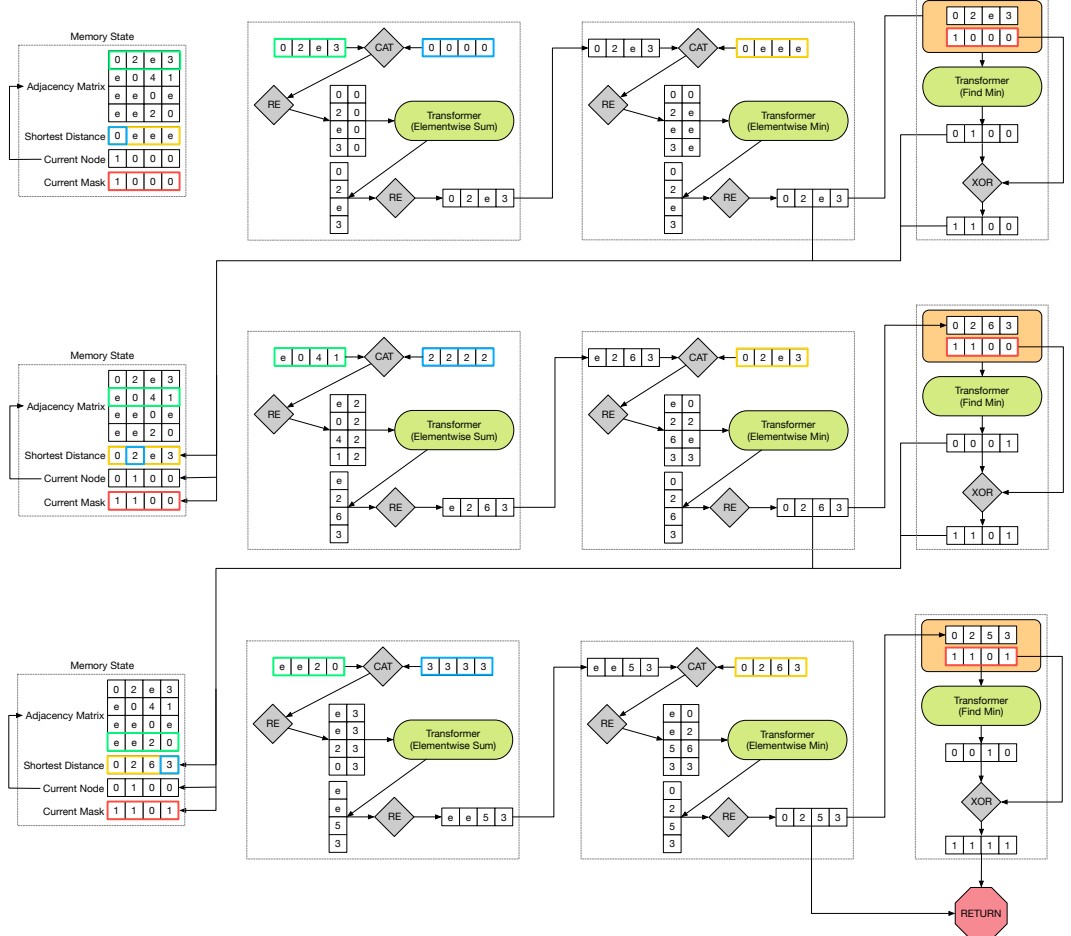

Figure 15: An execution trace of the Dijkstra algorithm for shortest paths with NEE components. Highlighted arrays and values correspond to referencing the memory inputs of the same corresponding color at that iteration.

