# OpenReview forum: "NEURAL EXECUTION ENGINES"
_ICLR.cc/2020/Conference — Reject_

### Official Review · AnonReviewer1 · 2019-10-22
**Official Blind Review #1**

**Rating:** 3

**Review:**

This paper investigates an interesting problem of building a program execution engine with neural networks. The authors proposed a transformer-based model to learn basic subroutines, such as comparison, find min, and addition, and apply them in several standard algorithms, such as sorting and Dijkstra’s.

Pros:
1. The method achieves generalization towards longer sequences than the sequences in the training set in several algorithms.
2. The method represents numbers in binary form and the visualization shows that it learns embeddings from fixed-range integer numbers in a well-structured manner.
3. The learned NEE subroutines are tested in a variety of standard algorithms, such as multiple sorting algorithms and Dijkstra’s shortest path algorithm. The experiments further demonstrate that several NEE subroutines can be composed together in complex algorithms.

Cons:
1. NEE mostly focuses on learning low-level subroutines such as number comparison or addition. Therefore, it has to be used along with conventional subroutines, and cannot completely replace the full execution in complex algorithms, which have sophisticated control logic, such as if/else and for-loops. When the transformer model is used alone in the sorting task (Sec. 4.1), the performance degrades substantially as the sequence length gets longer.
2. Although the method achieved some degree of strong generalization, it lacks a formal way to verify the correctness of the learned subroutines, as opposed to prior work on program synthesis (Cai et al. 2017) that can prove the generalization of their model with recursion.
3. The method relies on detailed execution traces for supervised learning which can be costly to obtain.

Questions:
1. Confusing sentence: “Can we retain high attention resolution by restricting the model to only observe the first scenario repeatedly?” Can you elaborate on what you meant here?
2. From Figure 1, it seems that the model with dot product attention generalizes better in longer sequences than the one with all modifications. What’s the reason?
3. I would like to better understand the limitation of these learned NEE subroutines in long sequences. For instance, in Figure 8 and Figure 9, how would the model perform beyond the lengths of the sequences tested here? Would the performance maintain at 100% or decrease gradually as the sequences get even longer?
4. I am curious to know how this method could be extended to support more complex number systems, such as float numbers, and more complex data structures beyond sequences, such as binary trees and priority queues. I’d love to hear what the authors have to say about this.
5. I'd also like to know if the number embeddings learn in different algorithms would exhibit different structures (by examining the visualization of number embeddings learned in different tasks).
6. As NEE focuses on learning the basic subroutines while NPI aims to learn the high-level program executions, I think that it’d be very interesting to see how these two can combine their complementary strengths to build a complete neural-based execution engine.

Typos:
Select he node --> Select the node


**Experience Assessment:**

I have published one or two papers in this area.

**Review Assessment: Checking Correctness Of Derivations And Theory:**

I carefully checked the derivations and theory.

**Review Assessment: Checking Correctness Of Experiments:**

I carefully checked the experiments.

**Review Assessment: Thoroughness In Paper Reading:**

I read the paper at least twice and used my best judgement in assessing the paper.

---

> ### Author Response · Authors · 2019-11-07
> **Review response**
>
> Thank you for your review, we very much appreciate your feedback.
>
> Response to cons:
>
> 1. We agree that this framework does not yet generalize to more complex subroutines, but we are able to handle both if/else statements (as in merge sort) and loops (selection sort, merge sort, Dijsktra) via the learned mask.
>
> 2. We agree that it would be interesting to try to formally prove strong generalization, or related properties of the model, however, we believe that this is beyond the scope of this work. Other neural network-based execution methods (e.g., NTM, Neural GPU, Learning to Execute) do not prove generalization formally, and instead leverage neural networks as universal function approximators. We believe that our work lies in a similar scope to these.
>
> 3. Depending on the program, these may or may not be expensive to obtain. However, this is an orthogonal issue for the time being. We sought to determine whether supervision was sufficient to improve strong generalization. This is similar to other related work using execution traces, such as neural programmer-interpreters. Reducing the level of supervision is a worthy goal, but we first sought to make sure that the problem is solvable with sufficient supervision.
>
> Response to questions:
>
> 1. We will improve the wording on this. We simply meant to posit whether “sharp” (low-entropy) attention values, as shown in Figure 2b, would persist when applied to longer sequences, or whether the entropy of the attention mask would increase with longer sequences (consequently degrading performance).
>
> 2. We don’t actually know the reason for this, although we did notice this trend. While dot-product attention does seem to perform better over longer sequences, it does still degrade. We can certainly investigate the cause of its improved robustness.
>
> 3. We haven’t tried very long sequences on merge sort or Dijkstra, but considering selection sort, we’ve tested out to length 500 sequences (training on length <= 8) and see no degradation in performance or degradation of other indirect metrics like the strength of the attention mask. At some point, we believe that performance would degrade, but we haven’t yet found it.
>
> 4. Other number formats (like floating point) or data structure types (like queues or heaps) are very interesting research directions. Our belief is that floating point in binary format could be more forgiving than pure integer, given the approximate nature of floating point precision and floating point applications.
>
> 5. Yes, there is definitely very different structure present in the learned embeddings depending on the task. One example is comparing Figure 13 (a) trained via sorting with Figure 13 (b) trained on addition.
>
> 6. Very much agreed, exploring how to combine NEE-like supervised approaches with more automatic NPI approaches and RL is a very interesting research direction.

---

### Official Review · AnonReviewer2 · 2019-10-23
**Official Blind Review #2**

**Rating:** 3

**Review:**

his paper deals with the problem of designing neural network architectures that can learn and implement general programs.  The authors are motivated by problems in such works, mainly generalization to testing distributions that do not necessarily correspond to the training distribution.  It should be made clear that the latter is a general problem in machine learning (with phenomena such as covariate shift being commonplace), the authors particularly relate this work to predicting new values and longer sequences (i.e. strong generalization).

The authors further motivate their work by assuming that these phenomena are due to lack of prior structure that can be alleviated by further supervision during training.  The goal is for complex behaviour to emerge via composition of simple functions (which clearly follows the deep learning paradigm).Specifically, the authors propose a modification to the transformer architecture, that does not use positional encodings (the authors mention that this was detrimental to their work - it would be good to provide some more insight into that) and single-headed attention.  The main contribution seems to be adding the self-attention mask that is learned, along with execution traces that have been used in previous work.  An relatively small increase in performance is observed due to this, but it seems that the experiment is limited (no standard deviation in results, so I presume one run with one initialization).  Therefore it seems to me that the contribution of this paper is limited in terms of technical contribution.

**Experience Assessment:**

I have read many papers in this area.

**Review Assessment: Checking Correctness Of Derivations And Theory:**

I assessed the sensibility of the derivations and theory.

**Review Assessment: Checking Correctness Of Experiments:**

I assessed the sensibility of the experiments.

**Review Assessment: Thoroughness In Paper Reading:**

I read the paper at least twice and used my best judgement in assessing the paper.

---

> ### Author Response · Authors · 2019-11-07
> **Review response**
>
> Thank you for taking the time to review our paper.
>
> We’re unclear why you say that there is a relatively small performance improvement. Specifically, we show that a standard seq2seq transformer can learn to sort lists of the same length on which it is trained, but completely fails at sorting longer lists (Figure 1: <10% accuracy at length 100, Figure 12: NEE achieves 100% at length 100). This is why we explored NEE - the resulting difference in performance in the longer regime is significant.
>
> Regarding standard deviations, while we did run the experiments many times and can verify that they are consistent, we agree that it is important to report error bars and are happy to do so.
>
> Positional Encodings: Sorting is a set operation, so the position of the numbers does not affect the result. Therefore, a positional encoding is not needed for sorting and potentially serves as a distraction for the model.

---

### Official Review · AnonReviewer3 · 2019-10-26
**Official Blind Review #3**

**Rating:** 1

**Review:**

# Summary

This paper trains a network to mimic simple known algorithms in a way that guarantees that they generalize to
out-of-distribution test instances. The network mimics the algorithms by running repeatedly in a loop where each
iteration of the loop runs a Transformer and outputs a mask that tells the next iteration the inputs to process. The
setup is tested on sorting, adding, and graph algorithms, and found to learn regular number representations that
supposedly aid generalization.

# Review

This paper has an admirable and useful goal, but the way it is currently implemented and presented is not ready for
publication at ICLR.

My main issue is with the training/testing setup and its presentation. The authors assume a certain structure of an
algorithm (for instance, the iterative structure of recursive selection sort), delegate one or more modules inside this
structure to be implemented by neural networks, and train them only.
Most of the "strong generalization" is coming from the fact that the iterative structure is fixed. The work abstracts
out the most complex parts of each algorithm. In Figure 3, for instance, the NN must learn to find the smallest element
among the non-masked-out ones on the input, return it, and mask it out. This is a much simpler task than the whole
sorting algorithm. Training the network to solve "find_min" != claiming that the network solves and strongly generalizes
on "sort".

Important training details are left unspecified. How is the data for training NEEs generated? For instance, for training
the network in Figure 3, do you trace the whole selection sort on a randomly generated list, and collect the
intermediate input/output pairs for "find_min"? If so, it's absolutely unsurprising that the process also works for
longer lists -- see above. Are composable NEEs, like the three networks in Figure 7, trained jointly or separately? Do
they observe their own outputs that are fed into subsequent NEE networks, or are the previous outputs teacher-forced,
or are they pre-trained? Many of these details need to be clarified precisely to make the experimental setup
verifiable.
Some important details are presented factually without any motivation. For example, why does Figure 5 use
SHIFT and XOR? Why, in general, the next mask produced by a NEE is XORed with a previous one instead of replacing it?

I liked the embedding visualizations, which clearly demonstrate structure in the latent space driven by (a) the
binary number representations, and (b) the addition task objective. In addition to regular ordering structure (needed to
implement addition), the latent space also clearly exhibits regularities inherent to the binary representation, such as
the shift by 64 in Figure 8a. While interesting, this only confirms the findings of Shi et al., albeit in a more pure
experimental setting.

In summary, the scope of experiments and presentation of results would need to be significantly improved in order for
this work to reach the quality bar of ICLR.

**Experience Assessment:**

I have published one or two papers in this area.

**Review Assessment: Checking Correctness Of Derivations And Theory:**

I carefully checked the derivations and theory.

**Review Assessment: Checking Correctness Of Experiments:**

I assessed the sensibility of the experiments.

**Review Assessment: Thoroughness In Paper Reading:**

I read the paper at least twice and used my best judgement in assessing the paper.

---

> ### Author Response · Authors · 2019-11-07
> **Review response**
>
> Thank you for taking the time to write a detailed review.
>
> The main contention appears to be that the generalization ability of the NEE is based on the iterative structure that we provide. In Section 4.1, we provide a baseline against this argument, and show that providing iterative structure does not guarantee strong generalization. The seq2seq view of the problem (Section 4.1) has the same iterative structure and input/output labels as the NEE version (Section 4.2). That is, both must repeatedly find the minimum, and both are provided supervision for the correct output at each step. The transformer does this through the decoder iteratively finding the next minimum value, while NEE does this through the conditional mask. Figures 11 vs. Figure 12 show that the transformer does not generalize as it learns a poor attention model. Do you have an alternative baseline to a seq2seq transformer that you believe would solve the problem?
>
> Iterative Structure: While we do target iterative algorithms, iterative structure is a natural property present in the source code of a large range of algorithms (recursive algorithms, map reduce, sorting, graph processing, compression, ...). There are many prior papers that target algorithms of fixed structure (Zaremba and Sutskever 2014, Cai et al. 2017).
>
> Simple Operations: We opted not to learn simple transformations that are independent of input data (like XOR), but can easily add these to the model since they are simpler than other data-dependent/learned operations (like addition). In this case, we use XOR as an illustration of a simple transformation (which could potentially be learned) as opposed to just replacing the mask, particularly since only a small fraction of the mask changes every iteration.
>
> Training Details: You are correct. In the case of selection sort we train NEE on execution traces (iteratively applying find_min) of random lists up to length 8. Note that the termination condition of selecting the end token is also learned, meaning NEE could stop early if it is not trained properly. For composing NEE modules, since we train with teacher forcing, each NEE is trained separately. You mention that you find it unsurprising that it works in this case, but a seq2seq model fails, even with the same training data. Again, if you have an alternative baseline we would be happy to consider it. We will modify the text to include more details about training.
>
> Embeddings: Shi et al. show that binary interpolates generalize better than other representations in a simple branching task. We show high performance in problems like addition, which are much more challenging and the subject of a significant amount of prior work (Zaremba and Sutskever 2014, Sutskever and Hinton 2009).

---

### Author Response · Authors · 2019-11-07
**A brief note to reviewers**

Thank you very much for your valuable feedback. We appreciate the time you took to review our paper, and believe that your comments will help us to greatly improve it.

Learning to execute general functions with neural networks is a very challenging problem. Our intention is to show that the standard end-to-end formulation (e.g., seq2seq transformers) fail to strongly generalize, even in simple cases with lots of supervision of intermediate states. The intention of NEE is to therefore explore strategies for overcoming these limitations. That is, we ask whether the current available machinery is sufficient to learn simple, yet challenging subroutines, given enough supervision and the right structure. Our intention with NEE is to show that this is indeed the case. As reviewer 1 points out in their questions, there are many interesting possible extensions to increase the complexity of the framework and associated subroutines.

All this is to say, rather than taking a top-down approach of trying to make the full end-to-end system with limited supervision work, we are instead taking a bottom-up approach and gradually expanding the scope of the framework in order to solve more challenging problems.

---

### Decision · Program_Chairs · 2019-12-19

**Decision:**

Reject

**Comment:**

This paper investigates the problem of building a program execution engine with neural networks. While the reviewers find this paper to contain interesting ideas, the technical contributions, scope of experiments, and the presentation of results would need to be significantly improved in order for this work to reach the quality bar of ICLR.